# Heterogeneity of Multiple System Atrophy: An Update

**DOI:** 10.3390/biomedicines10030599

**Published:** 2022-03-03

**Authors:** Kurt A. Jellinger

**Affiliations:** Institute of Clinical Neurobiology, Alberichgasse 5/13, A-1150 Vienna, Austria; kurt.jellinger@univie.ac.at; Tel.: +43-1-5266534; Fax: +43-1-5266534

**Keywords:** multiple system atrophy, atypical forms, clinico-pathological variants, MSA with lewy bodies, “minimal change” MSA, MSA with hippocampal pathology, conjugal MSA

## Abstract

Multiple system atrophy (MSA) is a fatal, rapidly progressing neurodegenerative disease of uncertain etiology, clinically characterized by various combinations of Levodopa unresponsive parkinsonism, cerebellar, autonomic and motor dysfunctions. The morphological hallmark of this α-synucleinopathy is the deposition of aberrant α-synuclein in both glia, mainly oligodendroglia (glial cytoplasmic inclusions /GCIs/) and neurons, associated with glioneuronal degeneration of the striatonigral, olivopontocerebellar and many other neuronal systems. Typical phenotypes are MSA with predominant parkinsonism (MSA-P) and a cerebellar variant (MSA-C) with olivocerebellar atrophy. However, MSA can present with a wider range of clinical and pathological features than previously thought. In addition to rare combined or “mixed” MSA, there is a broad spectrum of atypical MSA variants, such as those with a different age at onset and disease duration, “minimal change” or prodromal forms, MSA variants with Lewy body disease or severe hippocampal pathology, rare forms with an unusual tau pathology or spinal myoclonus, an increasing number of MSA cases with cognitive impairment/dementia, rare familial forms, and questionable conjugal MSA. These variants that do not fit into the current classification of MSA are a major challenge for the diagnosis of this unique proteinopathy. Although the clinical diagnostic accuracy and differential diagnosis of MSA have improved by using combined biomarkers, its distinction from clinically similar extrapyramidal disorders with other pathologies and etiologies may be difficult. These aspects should be taken into consideration when revising the current diagnostic criteria. This appears important given that disease-modifying treatment strategies for this hitherto incurable disorder are under investigation.

## 1. Introduction

Multiple system atrophy (MSA) is a rare and fatal adult-onset neurodegenerative disease of an uncertain etiology that has a variety of clinical and pathological variants. Belonging to the group of α-synucleinopathies, it is a unique proteinopathy with a specific glioneuronal degeneration involving striatonigral, olivopontocerebellar (OPC), autonomic and peripheral neuronal systems [1]. MSA is clinically characterized by various combinations of Levodopa unresponsive parkinsonism, autonomic dysfunctions, and cerebellar and pyramidal signs [2,3]. Its morphological hallmark is the abnormal accumulation of aberrant fibrillary α-synuclein (αSyn) in oligodendrocytes as glial cytoplasmic inclusions (GCIs) [4], as well as in select subsets of neurons [5] inducing cell dysfunction and death. The degeneration of specific neuronal pathways causes multifaceted clinical phenotypes: a parkinsonian variant (MSA-P), associated with striatonigral degeneration (SND), and a cerebellar variant (MSA-C) with olivopontocerebellar atrophy (OPCA) [2]. However, the clinical and morphological features of MSA are more widespread than previously considered and, in view of the limitations of the Unified Multiple System Atrophy Rating Scale (UMSARS), further development and validation of an impaired clinical outcome assessment is a priority for future research [6]. In addition to “mixed” MSA with a combination of both phenotypes, there are several clinico-pathological variants [1,7,8], which will be critically reviewed here in order to improve diagnostic accuracy of this disorder that shows overlap with other extrapyramidal disorders. The underlying pathogenic mechanisms are still not well understood, but there is evidence for the notion that a “prion-like” spreading of disease-specific αSyn strains is involved in the pathogenic cascade [9,10]. Together with oxidative stress, proteasomal and mitochondrial dysfunctions, myelin dysregulation, the redistribution of the oligodendrocyte-specific protein p25α, neuroinflammation, impaired neurotrophic factors, and energy failure, it is involved in the complex pathogenic process finally leading to a specific multisystem degeneration in this oligodendroglio-neuronal proteinopathy [1,9,11,12,13,14,15,16] (Figure 1).

## 2. ”Typical” MSA

MSA is a rare disease with an estimated incidence of 0.6–0.7 and a range of 0.1–2.4 cases/100,000 person–years [17]. Disease onset is 56 ± 9 years, with both sexes equally affected [18], but 20–75% of patients have a prodromal/preclinical phase with non-motor, mainly autonomic symptoms (cardiovascular, gastrointestinal, sexual or urogenital dysfunctions, orthostatic hypotension, or REM sleep behavior disorder /RBD/), which may precede motor presentation by years [19,20]. The mean survival from the onset of symptoms is 6–10 (mean 9.5) years, with only few patients surviving more than 15 years [21]. Parkinsonism with rigidity, bradykinesia, postural instability, a gait disorder tendency to fall and dysarthria dominates the motor presentation of MSA-P, while tremor is rare [22]. Hyperreflexia and Babinski signs occur in up to 50%, whereas abnormal postures like antecollis or dystonia are rare [23]. Cerebellar ataxia, widespread gait, uncoordinated limb movements, action tremor and spontaneous or gaze-invoked nystagmus predominate MSA-C [24]. These patients usually have a shorter delay with disease duration, RBD prevalence and cognitive assessment scores being similar, while the burden of autonomic symptoms is usually higher in MSA-P patients [25]. It should be emphasized, however, that about 50% of MSA-P patients develop cerebellar signs and even a higher proportion of MSA-C cases present parkinsonian features [26]. Respiratory disturbances, including inspiratory stridor and sleep apnea, are frequent [21] and are associated with a shortened lifespan and sudden death [20]. Dementia and hallucinations were previously considered as exclusion criteria for the diagnosis of MSA [2], but cognitive impairment has been increasingly reported in recent years [27,28]. 

The 18-FDG PET studies of patients with “probable” MSA revealed three different profiles, based on different brain metabolic activities: (1) extrapyramidal, axial, laryngeal-pharyngeal involvement (LPI) with cerebellar symptoms; (2) cerebellar and LPI symptoms; (3) cerebellar and cognitive symptoms [29]. 

The neuropathology of MSA-P shows frontal atrophy, severe atrophy and discoloration of the striatum, and depigmentation of the substantia nigra (SN) and locus ceruleus (LC), whereas MSA-C presents atrophy of the paleo- and neocerebellum, superior cerebellar peduncle, basis pontis and inferior olivary nucleus. Both phenotypes with a similar disease severity and duration have a differential distribution of gray and white matter atrophy, but white matter impairment is more severe in MSA than previously thought [30]. Rare Levodopa responders show a relatively slow progression in putaminal neurodegeneration [31]. Although cerebellar atrophy is a clear differentiation between groups, thalamic and basal ganglia structures are also relevant contributors to distinguish MSA subtypes [32]. The histological core features encompass (1) specific αSyn immunoreactive pathology with five types of inclusions: GCIs within oligodendroglia (Papp-Lantos bodies), which are mandatory for the postmortem diagnosis of MSA [4], and less frequent glial and neuronal nuclear inclusions (GCIs, neuronal cytoplasmic inclusions /NCIs), astroglial cytoplasmic inclusions, and neuronal threads [33]; (2) selective neuronal loss and axonal degeneration involving multiple regions of the nervous system; (3) widespread myelin degeneration with pallor, reduction of the myelin basic protein (MBP) and astrogliosis; and (4) extensive microglial activation and neuroinflammation with diffuse T-cell infiltration in the affected regions [34,35]. GCIs and the resulting neurodegeneration show a characteristic distribution, involving not only the striatonigral and OPC systems, but also multiple cortical regions, autonomic and motor nuclei in brainstem, as well as spinal cord autonomic nerve structures and the peripheral nervous system [1,36,37], characterizing MSA as a multi-system/-organ disorder [1,38].

## 3. Atypical MSA or Variants

Although all cases of MSA display neuronal loss in both striatonigral and olivopontocerebellar structures [39], with only 11 of 42 cases assigned in the category of “pure” MSA [40], this disorder has a wide range of clinical and pathological presentations, which expands the list of differential diagnoses [41]. Several subtypes do not fit into the current classification of MSA [42]: (1) MSA groups with different ages at onset and disease duration; (2) a “minimal change” or “preclinical” MSA as early pathological forms; (3) rare familial forms; (4) pathological variants, like MSA with Lewy body disease and with severe hippocampal atrophy or those with unusual tau pathology; (5) MSA with cognitive impairment/dementia; (6) MSA with dystonia and spinal myoclonus; (7) MSA-C with questionable conjugal MSA. 

The reasons for the clinical and morphological heterogeneity of MSA are a matter of current discussion. Possible pathogenic mechanisms could be the recently described diversity in αSyn species and differences in their seeding propensities in different brain regions from synucleinopathies [43], and the distinct biochemical properties of PD and MSA αSyn conformers, which is consistent with the idea that distinct αSyn strains underlie the heterogeneity among the synucleinopathies, e.g., PD and MSA [10,44]. Recent studies indicate that αSyn is the most enriched protein in PD and MSA extracts, which share a considerable overlap of their sarkosyl-insoluble protein, consisting of a vast majority of mitochondrial and synaptic proteins, while other fibrillary-prone protein candidates, possibly cross-seeded by αSyn, are neither found in PD nor MSA extracts. These results support the idea that pre-assembled building blocks originating in neurons are involved in the formation of GCIs in MSA and point to the sequestration of mitochondria and of neuronal synaptic components in both LBs and GCIs [45].

A comparison between young and aged mice injected with αSyn preformed fibrils (PFFs) showed that αSyn post-injection (PI) intervals rather than aging correlate with oligodendroglial αSyn deposition. These and other results provide a novel insight into the pathogenetic mechanisms of oligodendroglial αSyn aggregation in MSA [46]. Other recent studies showing a wider heterogeneity of αSyn seeding in MSA could explain a subclassification of MSA, which exceeds conventional clinical and neuropathological phenotyping and is related to the structural and biochemical heterogeneity of the accumulated αSyn [47]. These data support a model whereby the heterogeneity both between different brain regions and between different MSA patients might be due to differences in the cellular environment [48,49], and the presence of posttranslational modifications of αSyn [50] or other cofactors, such as p25α, an oligodendrocyte-specific protein [51], that may confer a selective pressure for one αSyn conformer over another [48]. Other studies have shown that the overexpression of αSyn by oligodendrocytes in transgenic mice does not recapitulate the fibrillary aggregation seen in GCIs in a human MSA brain, which could help to establish a link between αSyn aggregation in the development of a clinical phenotype of MSA [52].

## 4. Young-Onset MSA

Young-onset MSA (YOMSA) is rare. Among 455 patients, four (0.9%) developed the disease before the age of 40 [53]. Two presented with cerebellar symptoms, one of whom had a course typical of MSA, the other patient, after a rapid deterioration, died 3 years after disease onset. Two others presented with Levodopa-responsive parkinsonism, later developing motor fluctuations and peak-dose dyskinesias. Subthalamic brain stimulation resulted in the mild improvement of motor symptoms, but later deterioration occurred. There were no distinguishing clinical features for YOMSA in terms of presenting symptoms or survival, but a tendency to develop motor complications. Unfortunately, no autopsy data were available. Another group reported 22 patients with YOMSA, eight of whom with pathological confirmation [54]. These patients showed both clinical and pathological differences versus late-onset MSA (LOMSA), i.e., more common dystonia, Levodopa induced dyskinesias and pyramidal signs, whereas at postmortem analysis, the “minimal change variant” [55] was more common than in LOMSA, with a mean survival of 11.1 ± 3.2 (range 5.5–14.6) years). Among 44 autopsy-proven cases (24 MSA-P, 20 MSA-C), two showed disease onset before age 40. A female aged 33 years developed typical parkinsonism with a moderate transient Levodopa response, but progressive deterioration with autonomic failure, limb contractures, mild dementia and death in a decerebrate state 9 years after disease onset. Neuropathology revealed SND stage III [40] with generalized GCIs, occasional LBs but an intact OPC system. A male at age 39 developed orthostatic hypotension, mild cerebellar ataxia, increasing rigidity without tremor, bradykinesia, pyramidal signs and limb contractures, with death 9 years after disease onset. Neuropathology revealed SND and OPCA (both grade III), and a generalized GCI burden but no LBs; diagnosis was MSA-C+P associated with spino-cerebellar degeneration [56]. In comparison to the above YOMSA group, these patients presented with slow clinical progression and progressed pathological changes, suggesting various clinical and pathological phenotypes of YOMSA.

## 5. Later-Onset MSA (LOMSA)

Although in the consensus guidelines for the diagnosis of MSA [2], onset after age 75 is considered a non-supporting feature, some patients present with initial symptoms after age 75 [57,58]. Among 1425 clinically probable MSA patients in Korea, 39 had a mean onset at 76.8 years, with more frequent MSA-P (64.1%) than MSA-P in the mean-age at the onset group (40.6%). Dysautonomic symptoms, falls, limb ataxia and speech disturbance were more common in the LOMSA group. These patients had a shorter survival than the others. Among autopsy-proven cases, the SND dominant type was more common in the LOMSA group [57]. Among 171 autopsy-confirmed MSA cases in the Mayo Clinic brain bank (133 MSA-P, 35 MSA-C, 2 unclassified), six patients had LOMSA [58]. 

According to a recent study, LOMSA patients showed poor prognosis with a median survival time of 3 years, which is significantly shorter than in those with usual onset MSA (UOMSA) (4.8 vs 7.9 and 3.9 vs 7.5 years, respectively) [59]. Among 276 patients (193 in an autopsy cohort and 83 in a clinical cohort), LOMSA accounted for 8% and 5%, respectively. These and other studies showed that the clinical diagnosis of MSA may be difficult in elderly individuals due to the paucity of autonomic symptoms. All LOMSA patients presented with initial motor symptoms and had a more rapid disease progression than YOMSA patients, whereas no essential differences in the neuropathology of both cohorts were reported. Among 48 autopsy-confirmed cases of MSA (33 MSA-P and 15 MSA-C), with a mean age at disease onset of 55.5 ± 6.5 years, two females (both MSA-P) showed disease onset at age 75 and 80, respectively. The time from diagnosis to death was 2 and 4 years, respectively. In both patients the initial symptoms were gait disorders, bradykinesia and falls, without essential autonomic symptoms (orthostatic hypotension, urinary difficulties, etc.) or cerebellar signs. The clinical diagnosis was MSA-P, with the Hoehn & Yahr IV stage at the last visit. Both patients showed considerable cognitive impairment. Neuropathology showed SND grade III, associated with tau pathology akin to neuritic Braak stage III and V, respectively, and moderate cerebral amyloid angiopathy (CAA) [60]. In conclusion, these cases confirm previous observations showing that rare MSA patients with a late age of disease onset differ from the majority of MSA patients with a younger age of onset by mainly initial extrapyramidal syndromes without considerable autonomic symptoms and poorer prognosis, i.e., significantly shorter survival [61]. Cognitive impairment/dementia in these elderly MSA patients may be caused by considerable Alzheimer-related (and other) co-pathologies.

## 6. MSA with Prolonged Survival

MSA is a rapidly progressive disorder with a mean survival of 6–10 years, while 2–3% have a prolonged survival of 15 years and more [53,62]. Most of these patients had similar disease courses with a slow progression of parkinsonism in the first years and a subsequent rapid deterioration with the development of autonomic failure, making an accurate diagnosis of MSA difficult. Many of them developed motor fluctuations and Levodopa induced dyskinesias. Patients with MSA-P with slow progression and prolonged survival were considered as “benign” forms of MSA [62]. In a woman aged 82 years with a clinical course of 18 years, extensive distribution of GCIs along with NCIs involved the neocortex and limbic system. In particular, the granular cell layer of the dentate gyrus [63]. These findings were different from three MSA cases with disease durations of 15–19 years, showing the typical involvement of the striatonigral and OPC systems, but only a mild to moderate GCI burden without evident neuronal loss in the neocortex, hippocampus and amygdala [64]. A recently published case of MSA, with a 20-year disease duration, at autopsy showed severe gliosis and neuronal loss in a typical pattern, with atypical Pick body-like and ring-shaped αSyn inclusions, most prominent in limbic structures [65], consistent with a rare “atypical MSA” subtype. A Vietnamese woman aged 45 years, who died 21 years after her initial diagnosis of MSA-P, at autopsy showed both SND and OPCA with additional frontal and limbic atrophy, pathognomonic inclusions and Pick body and neurofibrillary tangle (NFT)-like αSyn-positive inclusions in the hippocampus, consistent with long-standing MSA-P with a limbic and frontotemporal lobar degeneration (FTLD)-type αSyn pathology [66]. An atypical case of FTLD-TDP type A with MSA phenocopy showed severe SND and cerebellar involvement [67], while four cases with clinical features of FTLD without autonomic dysfunction, presented frontotemporal atrophy and severe limbic αSyn neuronal pathology with Pick body-like but tau-negative inclusions. They were suggested representing a novel subtype of FTLD associated with αSyn pathology [66]. These observations indicate a highly variable involvement of different CNS areas in the rare cases of MSA with a prolonged clinical course.

## 7. ”Minimal Change” MSA

This aggressive form with GCIs and neurodegeneration almost restricted to the SN, putamen and LC, thus representing “pure” SND [68,69,70,71,72,73], indicates that GCI formation is an early event and may precede neuronal loss, thus being responsible for some of the clinical symptoms. One patient with preclinical MSA-C showed abundant GCIs in the pontine nuclei, middle cerebellar peduncle and cerebellar white matter, whereas NCIs restricted to the pontine basis, cerebellar vermis and inferior olivary nuclei were associated with neuronal loss, suggesting a link between both lesions in early stages of disease [74]. These patients presenting only mild clinical symptoms were recently suggested to represent “early MSA” rather than “minimal change” MSA [75]. Early MSA stages show a widespread increase of microglia (about 100%) in the white matter [76] without concomitant astrogliosis or essential oligodendroglial degeneration [77]. Both microglial activation and αSyn containing oligodendrocytes trigger neuroinflammation restricted to white matter regions with the preservation of gray matter areas [34]. A non-motor variant of pathologically confirmed MSA showed no overt parkinsonian or cerebellar signs [78]. The coexistence of the sporadic Creutzfeldt-Jakob disease with “minimal change” MSA was reported in a Spanish woman aged 64 [79]. A limbic counterpart of “minimal change” MSA with abundant αSyn pathology in the limbic system was reported in a male aged 70 with an onset of stiffness and swallowing difficulties at age 65. Based on symmetrical parkinsonism without tremor, dysphagia and autonomic failure, the clinical diagnosis was MSA. At autopsy, GCIs and NCIs were distributed throughout the brain with severe affection of limbic structures, especially the hippocampus, whereas the striatonigral and OPC systems showed only sparse inclusions. Therefore, the neuropathological diagnosis was “minimal change” MSA with limbic predominant αSyn [80]. This case implies not only the striatonigral or OPC, but also the limbic system can be predominantly involved in this rare subtype of MSA.

In neurologically normal individuals, GCIs are rarely found at autopsy as coincidental or incidental findings limited to the pons and inferior olivary nuclei with mild neuronal loss in SN, suggesting that these regions may be afflicted first in MSA-P [72,73]. In early MSA, the white matter microstructure was more affected than the gray matter. These changes were greater in MSA-C than in MSA-P, suggesting variable deterioration in the subtypes of MSA-C [81]. The presence of GCIs may represent an age-related phenomenon not necessarily progressing to overt neurological disease, classifying these cases as “prodromal/preclinical” or “incidental” MSA, similar to incidental LBD [82]. A recent data-driven classification of patients with early-stage MSA identified three subtypes. Almost half of them showed marked autonomic dysfunction and moderate parkinsonism, whereas the others presented with predominant parkinsonism or cerebellar symptoms but mild dysautonomia [83].

## 8. Familial MSA

MSA is generally considered a sporadic disorder, and a family history of ataxia or parkinsonism has been defined as a non-supporting feature in the current diagnostic criteria [2], but MSA pedigrees with both autosomal-dominant and autosomal-recessive inheritance have been observed in Europe and Asia [84,85,86,87,88], and there are reports of autopsy-proven MSA [89]. A genome-wide association study (GWAS) found an estimated heritability of 2–7% [90], but no single mutation linked to familial forms has been identified [24]. The link between V393A mutations and the *COQ2* gene, encoding the coenzyme Q10 (*COQ10*) and familial or sporadic MSA in Japanese and other Asian populations [91,92,93,94,95,96], has not been confirmed in other populations [90,97,98,99]. The *COQ2* V393A variant remains a susceptibility risk rather than causative, particularly for the MSA-C subtype in the east Asian population [100]. The *TBP* CAG/CAA repeat length of longer alleles (>38 repeats) is associated with an increased MSA risk, which supports a possible genetic overlap of MSA with spinocerebellar ataxia/SCA17 [101], causing the possibility of a misdiagnosis between MSA and SCAs [102]. Rare cases in a family with a pathological hexanucleotide repeat expansion in *C9orf72*, a gene linked to amyotrophic lateral sclerosis, presented clinical and neuroimaging features indistinguishable from MSA [103], and this gene may be involved in the heterogeneity of MSA [104].

## 9. MSA with Lewy Body Disease

Although MSA and Lewy body disease (LBD) are clinicopathologically distinct entities within the group of synucleinopathies, clinical presentations can sometimes overlap and some cases of LBD can be misdiagnosed as MSA [105,106], and both may have autonomic dysfunction and RBD. Furthermore, 10.7–22.7% of patients with MSA have concomitant LBD [8,106,107,108], but the combination of diffuse LBD and MSA has rarely been reported. Recently, 11 among 230 autopsy-proven MSA cases (5%) showed the characteristic clinicopathological features of MSA + LBD; seven were the brainstem type, three transitional and one the diffuse LBD type [109]. A case of FTLD-TDP type A with MSA phenocopy had an antemortem diagnosis of PD with dementia (PDD) or probable dementia with Lewy bodies (DLB). Two cases had neuronal loss in SN, but not in striatal or OPC systems, with widespread GCIs consistent with “minimal change” MSA. In these cases, LBD was considered the primary pathology, and MSA as coincidental. *APOE* ε4 allele frequency was not different between MSA patients with and without LBD, while 2/9 MSA + LBD patients had a risk variant of the GBA gene. Although rare, MSA with LBD can develop clinical features of PDD or DLB. “Minimal change” MSA could be interpreted as a coincidental but distinct synucleinopathy in a small subset of patients with diffuse LBD. Due to the fact that cognitive impairment/dementia and visual hallucinations occur in some patients with MSA [110,111], it may be difficult to make an exact differential diagnosis between DLB and MSA-P. There may be an overlap in the cognitive profiles of both disorders, although cognitive impairment is more profound in DLB [111], while fluctuating cognition, typical for DLB, appears to be absent in MSA [112]. However, it has been suggested that this feature may have been overlooked in MSA [113]. Although parkinsonism, autonomic dysfunction and RBD can be seen in both MSA and LBD, the presence of dementia and hallucinations, or dementia with RBD and parkinsonism should be considered as red flags for coexisting Lewy-related pathology [114].

## 10. MSA with Severe Hippocampal Atrophy

Although the striatonigral and OPC systems are the most vulnerable regions in MSA, abundant GCI burden and neuronal loss in other regions have been occasionally reported as pathological variants of MSA [66,115,116,117,118,119]. These patients may present with cortical symptoms, including frontotemporal dementia (FTD), corticobasal syndrome or others, which are not presented in classical cases of MSA [66]. Recently, among 146 autopsy-proven cases of MSA from Japanese clinics, 12 patients (8.2%; 7 MSA-P and 5 MSA-C) showed severe hippocampal pathology due to a heavy burden of αSyn immunoreactive NCIs associated with severe neuronal loss and astrogliosis in the hippocampal granular layer, CA1, subiculum, parahippocampal gyrus and amygdala [120]. The NCIs showed ring-shaped or NFT-like configurations, while three cases presented Pick body-like NCIs and severe atrophy of the medial temporal lobes with heavy NCI burden. In addition, LBs were seen in one of these cases, which showed Braak NFT stages I-III (mean 1.6 ± 0.8), being slightly higher than in the classical MSA. The patients with hippocampal MSA were younger at disease onset (mean 56.6 ± 9.3 vs 60.4 ± 9.3 years), had significantly longer disease duration (13.2 ± 5.9 vs 6.9 ± 3.0) and higher prevalence of cognitive impairment. This hippocampal subtype of MSA was considered a rare pathological variant and not the result of advanced disease. Among 48 autopsy-confirmed MSA (33 MSA-P and 15 MSA-C) with a mean age at disease onset of 55.5 ± 6.5 years and mean duration of 7.5 years, two females, both MSA-P and disease onset at age 61 and 75 and disease duration of 4 and 7 years, respectively, neuropathologically showed a considerable hippocampal pathology. Both patients presented initial rigidity, bradykinesia and gait disorders without tremor or essential autonomic failure. Clinical diagnoses were the PD rigid-akinesia type and MSA-P, respectively, with Hoehn & Yahr stage 4 to 5 at the last visit. The elder patient developed visual hallucinations, depression and moderate cognitive impairment, later laryngeal stridor and died of pneumonia. The younger lady, in addition to parkinsonian symptoms, developed mild muscular atrophies and a moderate cognitive impairment. Neuropathologically, both cases showed frontal and mediotemporal atrophy, SND grade III, with OPCA grade I in the elder one. In addition to multiple GCIs in the striatum, brainstem and less in the cerebral cortex, severe involvement by NCIs with neuronal loss and astrogliosis was seen predominantly in the hippocampal subarea CA1, granular cell layer, presubiculum, and parahippocampal gyrus. Neither Pick body-, nor NFT-like NCIs nor LBs were observed. In addition, moderate hippocampal tau pathology (Braak stage III) and mild Thal phases 0–2, but no tau-positive astroglia, cerebral amyloid angiopathy (CAA), TDP-43 co-pathologies or limbic and FTLD-type αSyn pathology were observed. The psychotic symptoms and cognitive impairment in both patients were correlated to the severe hippocampal involvement by αSyn and less by tau pathology. Neuronal mitochondrial dysfunction and altered ribostasis in NCI formation may be involved in the hippocampal degeneration of MSA [121].

## 11. MSA with Cognitive Impairment/Dementia

While cognitive impairment or dementia have been observed in 80–90% of patients with PD and DLB [1,122,123,124], they have been considered as exclusion criteria in the diagnosis of MSA [2]. However, a recent position statement by the Neuropathology Task Force of the Movement Society reported that cognitive impairment may occur in 17–47% of MSA patients, while severe dementia is rare [113]. Mild to moderate cognitive impairment has been reported in up to 40% in MSA-P patients and in 14.37% of autopsy-proven MSA cases [3,8,28,105,115,125], with a predominant impairment of executive functions and verbal memory [126,127]. In a retrospective study of 102 autopsy-confirmed cases of MSA, 33 (32%) had cognitive impairments, morphologically associated with a greater burden of NCIs in the limbic regions, in particular, in the dentate gyrus than patients without a cognitive impairment [8]. More severe and widespread cognitive dysfunction was seen in MSA-P than in MSA-C patients [128], while according to others, MSA-C patients performed significantly worse [126]. Cognitive impairments in MSA are probably due to involvement of prefrontal areas [129,130] causing striatofrontal dysfunction [131], or severe αSyn pathology in hippocampus [5,8,120,132]. Other structural changes in MSA-P patients with a cognitive impairment are cortical thinning, more severe in the fronto-temporal-parietal than in posterior brain regions, and a reduction of the subcortical gray structures [127], globular inclusion in the medial temporal region [115] or corpus callosum involvement [133]. However, others found no pathological differences between MSA patients with and without a cognitive impairment [134]. Only in a few autopsy-proven cases of MSA has ADNC been reported [3,135], while a clinical case of MSA with preexisting AD was observed [136]. Among 48 autopsy-proven cases of MSA (mean age at death 60.5 ± 7.8, range 46–82 years), mild to moderate cognitive impairment in 35.3% was associated with a moderate cortical tau pathology (Braak NFT stages II-IV) and cortical Lewy pathologies, while one had a probable primary age-related tauopathy (PART). A female aged 82 with a duration of 2 years and severe dementia showed fully developed AD (Braak stage V, ABC 3/3/3) and moderate CAA. LBs mainly in SN and LC were seen in 11 brains (20.9%) and in four of them, also, in the frontal cortex and dorsal nucleus of the vagus [27]. The GCI load in the striatum and the proportion of cases with subcortical small vessel disease did not significantly differ between MSA cases with and without dementia [107]. However, when combined with cerebrovascular risk factors and comorbidities, e.g., those causing autonomic failure, cerebrovascular pathology may masquerade as MSA [109,114]. Considering these findings in a limited number of MSA patients, the diagnosis of MSA requires the exclusion of other causes and further studies to elucidate the pathological basis of cognitive impairment in MSA are warranted [28].

## 12. MSA with Unusual Tau Pathology

A recent clinicopathological study of seven cases of MSA, in addition to characteristic morphological findings, demonstrated unusual tau-positive astroglia predominantly in the putamen, internal capsule, and pontine basis. These lesions were prominent in a female aged 72 who died 9 years after the onset of parkinsonism and ataxia [137]. The presence of tau-positive granules not co-localized with αSyn-positive GCIs in the oligodendroglia, with a more common expression of 4-repeat (R) than 3-R tau, and related to the severity of neurodegeneration in MSA, suggested that tau may be related to a neurodegenerative pathway different from that induced by αSyn, which is particularly associated with memory impairment in MSA [132].

Among 48 autopsy-confirmed MSA cases, only in one male aged 67, with 7 years duration of parkinsonism, in addition to SND grade III, tau-positive granules were detected by AT8 immunohistochemistry in the cytoplasm of astroglia in the degenerated putamen. These granules showed both R 3 and R 4 tau positivity, 4R tau being more common. Tau-positive granula were not co-localized with GCIs and double immunostaining showed no co-expression of tau and αSyn. No tau-positive glial granules were observed in any of the other 47 MSA cases, although AD-related lesions of a moderate degree were present in six cases of MSA aged 54–82 years [56]. A few other reports using the AT8 antibody have demonstrated that phosphorylated tau (p-tau) occurs in neurons, astrocytes, and oligodendrocytes of patients with a long disease duration [117,137,138] and atypical MSA cases with FTLD [66]. However, other p-tau sites, except for the epitope AT8 (p-tau 202/205), are less well studied in MSA. With antibodies against p-tau 231 instead of AT8, more numerous signals would be detectable in these MSA cases. The p-tau 231 expression level has been closely correlated with the Braak neuritic stages [139], and increased plasma p-tau 231 levels also correlate with amyloid-β pathology and prior to the threshold of amyloid-β PET positivity [140].

Two autopsy cases of MSA with tufted astrocyte-like glia (TUALG), a histopathological feature of progressive supranuclear palsy (PSP), a 4-R tauopathy, have been reported [125]. Clinically, both patients showed symptoms atypical of MSA, such as vertical gaze palsy. Neuropathology, in addition to MSA-typical changes, showed TUALGs in cerebral cortices but no other PSP tau pathologies. The coexistence of MSA and PSP is exceedingly rare. One patient with severe cerebellar ataxia, autonomic failure and rigid-akinetic parkinsonism, at autopsy, showed severe neuronal loss with gliosis in the putamen, SN, inferior olive and pontine basis, moderate neuronal loss and gliosis in the globus pallidus and subthalamic nucleus as well as NFTs and tufted astrocytes in the basal ganglia and brainstem. Double-labeling detected αSyn immunoreactivity in oligodendrocytes, phosphorylated tau in neurons and glia in the brainstem, indicating the coexistence of MSA and PSP [141]. Among 290 autopsy cases of PSP screened for αSyn and tau immunohistochemistry, a single case of PSP/MSA was detected. A female aged 86 years with clinical features consistent with PSP showed no signs of dysautonomia or cerebellar symptoms and imaging studies were not consistent with MSA. Neuropathology revealed tau-positive neuronal and glial lesions consistent with PSP as well as an αSyn-positive GCIs diagnostic of MSA. Double immunolabeling revealed no co-localization of αSyn and tau in most glial and neuronal inclusions [142]. Based on these findings, the neuropathological changes of PSP and MSA are distinct and independent processes, but they can occasionally coexist.

The pathogenic impact of tau accumulation in the astroglial cytoplasm is unclear, although links between αSyn and tau are suggested by the co-localization of both proteins in LBs and NFTs [143,144,145]. Whereas there are strong indications for an interaction between αSyn and tau in PD and other synucleinopathies [146,147], the relationship between these two proteins in specific cases of MSA awaits further elucidation.

## 13. MSA with Dystonia

In about 40% of untreated MSA patients anterocollis and unilateral limb dystonia represented the most frequent forms [148]. In five of 24 patients, four of whom with Levodopa responsive MSA-P, neuropathology confirmed the clinical diagnosis. A woman aged 57 with four years duration and unilateral end of dose focal dystonias, at autopsy revealed a minor neuronal depletion in the dorsolateral putamen with a heavy GCI load, severe focal neuron loss in the caudal ventrolateral substantia nigra compacta (SNc), and LC without LBs or pontine involvement, corresponding to MSA-P grade II [40]. In another patient with five years disease duration and craniocervical peak dose dystonia, neuropathology showed similar mild degeneration of the dorsolateral putamen and SNc, while two others with a disease duration of 4 and 5 years, respectively, and a generalized peak of dose dystonia, histopathologically showed subtotal degeneration of the dorsolateral and less of the anterior putamen and caudate nucleus, associated with mild gliosis of the globus pallidus and degeneration of SNc, particularly in the dorsolateral middle and caudal parts, corresponding to MSA-P grade III. A review of autopsy cases of MSA with dystonia in the literature and personal studies demonstrated that among eight cases with anterocollis, five were MSA-P and three MSA-C. All the former were characterized by severe degeneration of the putamen, particularly involving the dorsolateral part, and the SNc. Among 28 cases with generalized spastic dystonia, 21 were MSA-P and 8 MSA-C; 26 of them showed severe degeneration of the putamen, 12 of the caudate nucleus, seven of the globus pallidus, and in all except for one, severe involvement of the SNc. Only single cases showed involvement of the thalamus, corpus subthalamicum and other brainstem nuclei, such as the LC, substantia nigra reticulata, arcuate nucleus, and others. These data demonstrated that dystonia is more frequent in MSA-P than in MSA-C; in most but not in all cases it is associated with the severe degeneration of the dorsolateral putamen and SNc, and less of the ventral putamen. While in MSA with spastic dystonia, both the putamen and SNc appear equally involved, in those with dystonic contractures, the putamen is usually more often affected than the SNc, whereas affection of other subcortical brainstem nuclei and the pontocerebellar system is very rare. This conforms to previous studies on focal dystonia emphasizing the role of the putamen as a major site of this particular dysfunction [149,150], that is related to impairment of the basal ganglia circuitry [151]. However, in view of the variability in the intensity and distribution of the morphological lesions in MSA with dystonia, their pathophysiological mechanisms need further elucidation [152].

## 14. MSA with Spinal Myoclonus

Myoclonus has been reported in MSA with a prevalence ranging from 16.6% to 95% [153,154,155], but its mechanism is not well understood. Among 22 cases of definite MSA-C, three showed myoclonus between one and 6.17 years from symptom onset. Immunohistochemical studies revealed more αSyn deposition in motor-related regions of the spinal cord compared to those without myoclonus [156]. A male aged 66 presenting with dysarthria, ataxia, falls, autonomic dysfunctions and multifocal startle myoclonus died after five years. Neuropathology, in addition to the degeneration of the cerebellum, pontine basis, middle cerebellar peduncles, spinocerebellar and lateral spinal tracts, revealed an abundant αSyn deposition in oligodendroglia in the gray matter of the spinal cord (anterior and posterior horns), and white matter tracts. This case also appears to represent spinal myoclonus in MSA-C caused by the deposition of αSyn in the spinal cord [156], but larger studies will be necessary for understanding the neurobiology of myoclonus in patients with MSA.

## 15. Conjugal MSA

PD in one spouse and MSA in the other have been reported, while a recent paper described a married couple in which both husband and wife developed MSA symptoms at age 63 years. The husband, after developing parkinsonism and global autonomic failure, died at age 66, two years after symptom onset. The wife, at age 63, developed urinary incontinence, dysarthria, dysphonia, sleep apnea, parkinsonism and action tremor within one year. It was suggested that this case of conjugal MSA likely occurred by chance, although exposure to shared risk factors (pesticides) could not be excluded [157]. Another case of conjugal MSA-C was reported in a Japanese couple [158]. The wife at age 54 presented with finger movement disorders followed by gait disturbance, postural instability, obstipation, adiadochokinesia, RBD, hyperreflexia, and orthostatic hypotension. A brain CT revealed cerebellar and pontine atrophy indicating probable MSA-C. Two years after her disease onset, the husband, aged 58, presented with gait disturbances, adiadochokinesia and orthostatic hypotension. A brain MRI showed cerebellar and brainstem atrophy, a typical hot cross sign in the pons; SPECT revealed hypoperfusion of the brainstem and cerebellum, suggesting the same diagnosis. The couple denied any environmental risk factors, and genome sequencing excluded any mutations typical for MSA or spinocerebellar ataxia, but evaluation of the genetic risk factors of MSA suggested that the couple, in particular the wife, showed risk alleles in all single nucleotide polymorphism examined. These and other examinations provided no evidence to exclude the possibility of person-to-person transmission. Both these studies of clinically probable conjugal MSA may have important implications for conjugal α-synucleinopathies. Even though the transmissibility of MSA in a prion-like pattern is an intriguing hypothesis, considering both the investigated environmental as well as genetic risk factors, they do not allow any definite conclusions in this regard [159].

## 16. Conclusions and Outlook

MSA is a rapid progressive and fatal neurodegenerative disorder with currently no available effective treatment, but its accurate diagnosis is important for the proper management of patients, particularly in the early course of the disease. However, due to the great variability of the clinical presentation based on the neuropathological heterogeneity of MSA, the current clinical diagnostic criteria have some limitations not only in early, but even in progressed stages of the disease, since they do not include recent clinical and laboratory findings which may improve the diagnosis, and do not capture the large number of clinical and morphological variants of MSA [160,161,162,163,164,165]. These aspects should be taken into consideration when revising the current diagnostic criteria of MSA [41,166]. A critical review of 218 MSA cases showed that 177 (81.2%) were clinically diagnosed and pathologically confirmed as MSA (i.e., typical cases), while the remaining 41 (18.8%) had received alternative clinical diagnoses, including PD (*n* = 16) and PSP (*n* = 17) [167]. It is actually the high rate of disease progression that changes the initial diagnosis of usually PD or other movement disorders to MSA at later clinical visits, while the diagnostic difficulties in prodromal or initial disease stages are much higher [168,169,170]. Given the modern disease-modifying treatment procedures for MSA being under investigation [160], recognition of an MSA patient in an early phase of the disease as much as possible is urgently warranted.

## Figures and Tables

**Figure 1 biomedicines-10-00599-f001:**
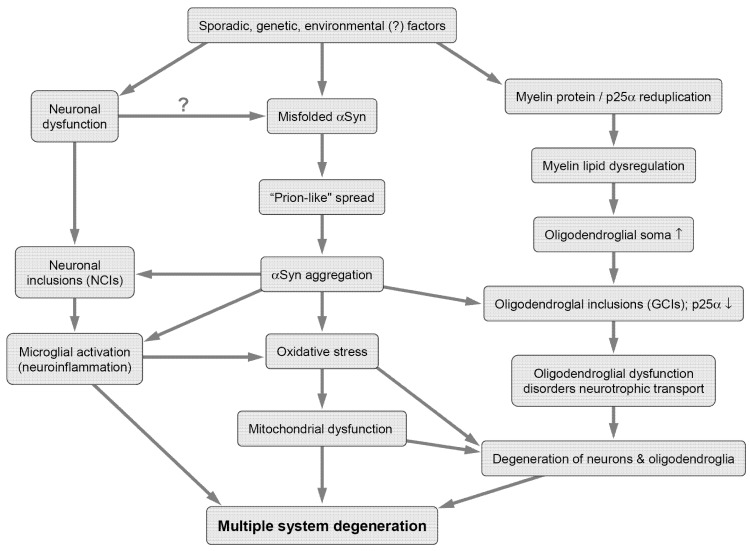
Putative pathogenic pathways of multiple system atrophy. NCIs: neuronal cytoplasmic inclusions; GCIs: glial cytoplasmic inclusions; αSyn: α-synuclein.

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
