# Peer review of "Heterogeneity of Multiple System Atrophy: An Update"

_biomedicines, 2022, doi:10.3390/biomedicines10030599_

Round 1
Reviewer 1 Report
I particulalry enjoyed reading this review by Kurt Jellinger where he points to the heterogeineity of MSA. This is a very detailed account of the clinico-pathological presentations of the disease and will without doubts become a reference article in the field. I only regret that a few very recent reports that seem quite relevant to MSA heterogeneity and pathophysiology are not referenced in this update:
- two of them deal with aSyn seeding heterogeneity in MSA (DOI: 10.1186/s40035-022-00283-4) and in PD and DLB (DOI: 10.3390/biom11060820). These observations are certainly relevant to the notion of clinical heterogeneity.
- three other papers point to the possibility that the oligodendrogial pathology seen in MSA is a distal feature due to the transfer and the storage of pathological assemblies first assembled in neurons (DOIs: 10.1093/jnen/nlz070 ; 10.3390/cells9112371 ; 10.1038/s41531-021-00264-w). This notion might be relevant to the origin, the mechanism and the heterogeneity of MSA as well.
As per the title of the MS that annouces an update, I would thus suggest to include/discuss these recent observations.
Author Response
Many thanks for your kind review and the proposals concerning the remarks about the heterogeneity of MSA phenotypes, which have now been discussed in a special part about differences in αSyn seeding and other relevant mechanisms. In addition, the detection of p-tau in MSA has been further discussed.
Reviewer 2 Report
The article by Jellinger provide comprehensive overview of the heterogeneity of multiple system atrophy, a rare and fatal adult-onset neurodegenerative disorder of uncertain etiology. The review is especially important since MSA share clinical symptoms with several other α-synucleinopathies, therefore making diagnosis challenging.
Regarding to that, it will be worth to comment that the number of misdiagnosed patients is actually much larger in initial disease stages than stated in manuscript conclusions “The critical review of 218 MSA cases showed that 177 (81.2%) were clinically diagnosed and pathologically confirmed as MSA (i.e., typical cases), while the remaining 41 (18.8%) had received alternative clinical diagnoses, including PD (n=16) and PSP (n=17)“. It is actually high rate of progression what change the initial diagnosis, of usually PD, to MSA at later clinical visits.
Beside, only small spelling and punctuation mistakes should be corrected.
Author Response
Thanks for your review and the comments about diagnostic difficulties in prodromal stages of MSA, which have been shortly discussed at the end of the conclusion chapter.